# Sound Localization Ability in Dogs

**DOI:** 10.3390/vetsci9110619

**Published:** 2022-11-08

**Authors:** Cécile Guérineau, Miina Lõoke, Anna Broseghini, Giulio Dehesh, Paolo Mongillo, Lieta Marinelli

**Affiliations:** 1Laboratory of Applied Ethology, Dipartimento di Biomedicina Comparata e Alimentazione, University of Padova, Viale dell’Università 16, 35020 Legnaro, PD, Italy; 2Independent Researcher, Via Chiesanuova 139, 35136 Padova, PD, Italy

**Keywords:** dog, minimum audible angle, staircase method, learning

## Abstract

**Simple Summary:**

An animal’s ability to localize the source of a sound on the horizontal plane is commonly measured by the minimum audible angle (MAA), i.e., the minimum angular distance between two possible sources at which an animal is still able to identify which of the two produced a sound. Only two studies explored this parameter in dogs, and both were characterized by relevant limitations: a very small sample size, and reliance on a method of assessment that might have not produced reliable results. To overcome the latter limit, we used a staircase method, whereby the dog’s rough threshold was quickly reached, and then several tests were performed around the actual perceptual limit of the dog, allowing for greater precision in estimation of the actual threshold. Moreover, the assessment lasted until the dog’s performance showed a clear stabilization. Ten dogs completed the experiment, testing angles between 60° and 1°. Their MAA ranged between 1.3° and 13.2°, with an average of 7.6°. Dogs also showed an improvement in performance across the procedure. The results show that the staircase method is feasible and reliable for assessing sound-localization sensitivity in dogs. They also reveal that the MAA of dogs is more variable than previously reported, potentially reaching values lower than 2°. Larger-scale studies should be performed to explore if and how factors such as ear and head shape or age influence sound localization abilities in dogs.

**Abstract:**

The minimum audible angle (MAA), defined as the smallest detectable difference between the azimuths of two identical sources of sound, is a standard measure of spatial auditory acuity in animals. Few studies have explored the MAA of dogs, using methods that do not allow potential improvement throughout the assessment, and with a very small number of dog(s) assessed. To overcome these limits, we adopted a staircase method on 10 dogs, using a two-forced choice procedure with two sound sources, testing angles of separation from 60° to 1°. The staircase method permits the level of difficulty for each dog to be continuously adapted and allows for the observation of improvement over time. The dogs’ average MAA was 7.6°, although with a large interindividual variability, ranging from 1.3° to 13.2°. A global improvement was observed across the procedure, substantiated by a gradual lowering of the MAA and of choice latency across sessions. The results indicate that the staircase method is feasible and reliable in the assessment of auditory spatial localization in dogs, highlighting the importance of using an appropriate method in a sensory discrimination task, so as to allow improvement over time. The results also reveal that the MAA of dogs is more variable than previously reported, potentially reaching values lower than 2°. Although no clear patterns of association emerged between MAA and dogs’ characteristics such as ear shape, head shape or age, the results suggest the value of conducting larger-scale studies to determine whether these or other factors influence sound localization abilities in dogs.

## 1. Introduction

Spatial hearing allows a listener to identify the position of sound sources [1] and this ability is crucial for survival in the animal kingdom. In order to localize the source of a sound, the azimuthal angle (horizontal plane), the vertical angle (vertical plane) and the distance (depth) need to be evaluated by the listener [2]. Terrestrial vertebrates developed spaced auditive sensory structures in the two ears which allow them to encode spatial information about a sound and localize its source [3]. Interaural differences in time and level of the perceived sound are used by mammals to localize sound sources on the horizontal plane [2]. Interaural time difference (ITD) is the time needed by a sound to propagate between the two ears, and is mostly effective in locating low frequency sound (below 1500 Hz in humans). In contrast, the interaural level difference (ILD) is the difference in sound intensity between the two ears, allowing location especially of higher frequency sounds (more than 3000 Hz in humans). Several studies have shown that the accuracy of location on the horizontal plane is dependent on the bandwidth [4], being higher for wideband noise stimuli than for narrow-band noise or pure-tone sound [5]. In addition, head movement during the presentation of the stimulus leads to poorer accuracy than if the head is immobile [6] although opposite effects have also been reported [7]. Sound localization can be enhanced in humans by rotating the head on the azimuthal plane (i.e., left–right), which helps maximize binaural differences. In animals with movable ear pinnae, it is likely that some contribution to sound localization is also provided by movement of the external ear. For instance, in cats the accuracy of sound localization was identical regardless of whether the head was stable or not, suggesting that movement of the pinnae compensated for the absence of head rotation [8]. It is possible that the same benefit would be observed in dogs, especially in breeds with erect ears and with meaningful sounds [9].

The intensity and duration [10], the amplitude modulation [11] and the elevation of the sound [12] do not seem to affect the accuracy of sound-source localization in free- field conditions. Accordingly, the ability to localize a sound with highest accuracy is provided by sounds with a large bandwidth, so that the listener can effectively rely on both ITD and ILD.

In order to evaluate the ability of human and non-human animals to localize a sound in their environment, two main types of procedures have been developed. The first, referred as the absolute procedure, measures the actual ability of a subject to indicate the location in space of a single auditory stimulus (by pointing to where s/he thinks the sound comes from). The second is the relative procedure, and measures the subject’s ability to determine either the final location of a repeated auditory stimulus that has changed location or the smallest detectable difference between the azimuths of two identical sources of sound, that is the minimum audible angle (MAA) [13]. As it is difficult for non-primates to point to where the sound source is perceived (absolute procedure); in these species the relative procedure is used, where the animal is required to discriminate if the sound comes from left or right on a given azimuth to determine the MAA [14].

To the best of our knowledge, only two studies have used behavioral methods to assess sound-localization capability in dogs. Specifically, MAA was assessed in three dogs with a two-choice test using pure-tone sounds [15], and in one dog using noise burst [16]. The MAAs found in these studies were 7° and 8°, respectively. The evident limit of both articles is the small number of tested subjects, which greatly prevents the generalization of the results. Indeed, intra- and inter-variability of results among dogs were impossible to assess or were not assessed in those studies. In addition to sample numerosity, the methods used to assess the MAA of dogs may not have allowed an accurate assessment. Heffner and Heffner [16] used an approach akin the method of limits [17], ending the evaluation when the animal could no longer distinguish if the sound came from left or right. Babushina and Polyakov [15] used a slightly modified method of constant stimuli [17], testing predetermined angles of separation of the emitters. The main shortcoming of those methods is that most data are obtained far away from and not focused on the region of interest (i.e., MAA). Moreover, with such procedures it was impossible to detect potential improvement of dogs’ capability during the evaluation, leaving open the possibility that the actual MAA was lower than the one reported. 

To overcome these limits, we adopted a staircase procedure [17]. In this methodology the angle of separation of the emitters was increased or decreased on the basis of the previous performance of the animal, making sure that most of the data collected was around the MAA. Moreover, descending and ascending assessments were performed for each subject. In the descending assessment the angle of separation of the first trial was well above the presumed threshold and the subject needed to respond to progressively lower levels of separation, while the ascending assessment started with an angle of separation under the hypothetic threshold with progressive increments. As the subjects’ performance in a sensory discrimination may change, depending on whether trials progress from easy-to-hard- trials or vice versa, using both approaches allows for a more complete evaluation. In addition, performing two assessments allowed us to see potential improvements, as sound localization capability may improve by repeated exposure [18,19]. Therefore, our aim was to assess the feasibility of a procedure based on the well-known psychophysical staircase method, in order to study sound localization abilities in dogs, overcoming the limits of previously adopted methods. 

## 2. Methods

The MAA was determined on the horizontal plane for ten dogs in a two-choice-test procedure using a white-noise sound.

### 2.1. Subjects

The sampling method was opportunistic, with the inclusion criteria for the dogs to be in a good health without known hearing impairments, to have a willingness to cooperate in the laboratory setting and to be food-motivated. Ages were between 1.0 and 7.9, to minimize age effects of youthful exuberance and potential age-related hearing alterations [20]. As the experiment required the repeated presence of the dog at the laboratory facility over an extended period, dogs were also recruited based on their owner’s availability. All owners were part of the University of Padova staff or students.

The final sample was composed of ten dogs (seven females and three males), 4.0 ± 2.5 years old. The interaural distance was 11.6 ± 1.9 cm. The interaural distances were collected through a caliper positioned at the ear area, under the entrance of the ear canal (for more details regarding individual dogs’ characteristics, see Table 1).

### 2.2. Experimental Setting

Dogs were tested in a room (5.8 m × 4.7 m), which had been set up to attenuate outside noise and reduce sound reflection. For this purpose, the walls were covered with heavy curtains, carpets were placed around the apparatus and sound-absorbing panels (model B, Zstyle, Corno di Rosazzo, Italy) were placed in front of the animal, behind the speakers, on the ceiling and on the windows.

The spatial configuration of the experimental setting is depicted in Figure 1 and the apparatus is depicted in Figure 2. A wooden board, covered with a soft bedding for comfort, was placed at the center of the apparatus, at a height which varied according to the height of the dog’s head (from 47.0 to 72.5 cm), so that the dog could naturally rest its head on it. Wire-mesh walls were placed at both sides of headrest, and their width could be adjusted in accordance with the width of the dog’s head, so to ensure that the dog’s head on the headrest was straight. An operator sat at about 1.5 m in front of the apparatus, concealed from the dog’s view by a plastic, opaque panel (100 cm wide, 120 cm tall), covered with sound-absorbing material. A lateral mirror allowed the sound operator to see the dog on the headrest and a food dispenser (Treat & Train, Premier ^®^, PetSafe, Knoxville, United States) was placed 132.5 cm from the dog’s position. At 45 cm in front of each food dispenser, a line was placed on the floor, which would be used to define the moment the dog made a choice (see procedure and data collection for further details). Aside each food dispenser, one custom-built speaker (later referred to as the training speaker) was placed on the floor in a fixed position, at 120° of separation and 148 cm distance behind the headrest and turned towards it. These speakers were powered by a custom-built amplifier, based on a TA2024 amplifier chip. Behind the apparatus, two active speakers (Inspire T6100, Creative Labs, Inc. Milpitas, California, USA), later referred as the testing speakers, were positioned according to the desired angle of separation, between 60° and 1° of separation, and at a distance between 300 and 310 cm behind the headrest, and oriented towards it. The position of the different speakers referring to a specific angle of separation was calculated between a line connecting the speaker and the dog’s head-position on the headrest, and the sagittal line.

A second experimenter, the dog operator, was located between the two testing speakers about 3 m behind the dog. Three cameras (two WV-CS570 and one WV-CP310, Panasonic, Delhi, India) were used to record all the sessions. One camera was focused on the operator’s area, to determine exactly when the sound was played, one was focused on the position of the dog’s head in the apparatus, and one captured the entire width of the room, with the two food dispensers, and was used to determine the side and timing of the dog’s choice.

### 2.3. Acoustic Stimuli

The same acoustic stimulus was used throughout all phases of the study, i.e., a white noise with a duration of 750 ms (with 250 ms fade-in and 250 ms fade-out effects to avoid switch-transient phenomena). The sound was generated with the Audacity^®^ software [21] and reproduced during the experimental procedures by a MacBook Pro computer (13 inch, Mid. 2012, Apple Inc., Cupertino, CA, USA), which sent the sound to the appropriate speaker.

A sonometer (2250-S, Brüel and Kjaer, equipped with microphone Model Brüel and Kjaer 4144, Naerum, Denmark) was used to ensure that the sound pressure at the headrest was constant and equal to 65 dB SPL so to avoid the possibility of the dog discriminating speakers by cues other than the spatial location.

### 2.4. Experimental Procedure

Initially, some time was spent by the dog operator with the dog outside of the laboratory, in order for the dog to be gain acquaintance with the operator. This phase usually lasted half a day. When the dog seemed at ease with the operator, it was taken to the laboratory for the next phase. Here, a preliminary training was performed to teach the dog to place its head on the headrest and to wait in a standing position. This was achieved by shaping, aided by a clicker as secondary reinforcement. The dog was also habituated to the automatic operation of the food dispenser and trained to reach the food dispenser beside the speaker where the sound was produced. When these two procedures were learnt, both were combined and the dog needed to position its head on the wooden board, wait for the production of the sound by one of the two training speakers, and reach the food dispenser beside the speaker where the sound came from, to obtain a food reward.

When the preliminary training was completed, the experimental phase began. The dog operator stayed behind the dog during the entire session and asked the dog to place their head on the headrest using a gesture and/or verbal cue. When this occurred, a sound was played once by the right or left speaker. If, after hearing the sound, the dog approached the food dispenser located beside the sound source (i.e., correct choice), it was rewarded with a dog kibble provided by the food dispenser in addition to verbal praise (“Good dog!”) given by the dog operator. Otherwise, the latter said “No!” and recalled the dog to the starting position. 

Sessions were composed of 14 trials. Initially, only the training speakers, at 120° of separation, were used. Once the dog was able to discriminate the sound source in two consecutive sessions, with two or fewer mistakes, the procedure was generalized by introducing the test speakers at 60° of separation, in 6 out the 14 trials. To complete this training phase, the dog needed to perform two consecutive sessions, with a maximum of one mistake in the 60° trials. 

Once the learning criterion in the training phase had been achieved, the test phase began. Test sessions were composed of a sequence of 14 trials, 8 of which were training trials and 6 test trials. The angle of separation of speakers in training trials was always 120°. The angle of separation of speakers in test trials varied from 30° to 1°. The angle of separation used in test trials varied between but not within sessions, so that the same angle was used in all test trials of a session. The sequence always initiated with two training trials, followed by alternation of one test and one training trial. The direction where the sound came from in each trial was semi-randomized, with no more than three trials in a row from the same side, and sides balanced within both the training and the test trials. Both test and training trials were always rewarded with dog kibble and verbal praise if the choice was correct. Training trials were included to prevent frustration and to verify that the dog was performing with adequate attention/motivation: if within the eight training trials, the dog failed two or more times, the entire session was considered invalid and data obtained in that session were eliminated from the final dataset. Sessions were separated by a minimum of 20 min with no more than six sessions performed by each dog in half a day.

The procedure described above was used for two subsequent assessments, named as the descending and ascending assessment, in this order. In both assessments, progression from one session (i.e., a specific angle of separation) to the next was based on the dog’s performance: if within the six test trials of the session the dog failed on less than two trials, it was considered as a success and the following session was done with a lower angle of separation. If it failed two or more times in test trials, the test session was considered as failed and the subsequent session was done with a higher angle of separation. 

The descending assessment started with the establishment of a rough threshold, by placing speakers at 30° of separation, then 15°, then 10°. If the dog failed at 30°, 15°, or 10°, then the increment in angle of separation for the subsequent sessions was done in steps of 3°, until the dog succeeded. The angle at which the dog succeeded, or 10° if the dog had succeeded at that angle on its first attempt at this level, was considered the starting point for the actual staircase procedure. Starting from this value, all subsequent increments or decrements in angle of separation were done in steps of 1°. Sessions in which the performance of the dog reversed compared to the previous sessions (i.e., the first session in which the dog responded incorrectly after a correct session, or the first session in which the dog responded correctly after an incorrect session), were considered as reversal sessions. Sessions proceeded until a minimum of 10 reversal points were observed and the last six reversals were within a range of 3° from the lowest to the highest in the series. 

Once the descending assessment was completed, the dog underwent the ascending assessment, which started from an angle of separation which was, for any given dog, 1° lower than the average of the six previous reversals of the descending assessment for that dog. The staircase method started from the very first session and increments or decrements in angle of separation were done in 1° steps from the very beginning. All other procedural aspects were identical to those described for the descending assessment.

If, at any moment during training or testing, the dog showed signs of distress, the session was interrupted and redone at a later time, when the dog was again at ease. If the dog showed any persistent signs of distress, the experiment was interrupted on that dog and the latter would have been excluded from the study. None of the previously described situations occurred with any of the dogs in this experiment.

### 2.5. Data Collection and Analysis

The average angle of the previous six reversal sessions of both the descending and the ascending assessment was calculated. The latter was considered as the MAA for any given dog. 

Behavioral data were collected from videos using Observer XT software (version 12.5, Noldus, Groeningen, The Netherlands). A continuous focal animal-sampling procedure was used. For each trial, we collected the tested angle of separation of the speakers, the exact moment and the side the sound was produced, the moment when the dog crossed the line defining the choice and if the dog-choice was correct. 

Collected data were used to calculate latency from the moment the sound was produced to the moment the dog made a choice. For the analyses, the mean latency of test trials and of training trials was calculated for each reversal session.

To assess inter-observer reliability, a second coder collected data from 25% of the 420 videos (chosen so that all dogs and failure and success sessions were equally represented). The collected data were compared using the intraclass correlation coefficient, resulting in an inter-observer reliability between 0.88 and 0.99.

In order to analyze how the dog performance changed across the entire procedure, the reversal sessions were grouped in batches. Four different batches were created, with batch B1 including the first four reversals of the descending assessment, B2 the last four reversals of the descending assessment, B3 the first four reversals of the ascending assessment and B4 the last four reversals of the ascending assessment. 

In order to determine whether there was an improvement in the dogs’ sound localization ability across the procedure, we used a generalized estimating equations (GEE) model. The dependent variable was the angle of separation of the speakers. The model included the batch as a fixed factor and the dog’s name as the random factor accounting for the repeated measurement taken from the same dog. Post hoc pairwise comparisons were performed if a significant effect was found for the batch. 

A GEE model was also used to determine whether there was any change of latency across the procedure and whether the latency depended on the difficulty of the task. The model included mean latency as the dependent variable. It included batch, type of trial where the latencies were taken from (training/testing), type of reversal (success/failure) and first- and second-level interactions, with the dog’s name as the random term. The backwards elimination procedure was applied to obtain the final model. Post hoc pairwise comparisons were performed if a significant effect was found for any interaction or model term. 

## 3. Results

In the descending procedure, dogs performed, on average, 25.1 ± 4.5 sessions, of which 11.1 ± 1.6 were reversal sessions. In the ascending procedure dogs performed, on average, 19.2 ± 5.7 sessions of which 10.8 ± 2.5 were reversal sessions.

The mean MAA in our sample was 7.6 ± 3.4°, ranging from 1.3 ± 0.5 to 13.2 ± 1.2° (see Appendix A in Appendix A for individual details).

Figure 3 shows the mean ± SD of the angle of separation in different trial batches of both the descending and ascending assessments.

The mean angle of separation in the batch at the beginning of the descending assessment was higher than in batch at the end of it (B1 = 11.8 ± 1.1°, B2 = 9.6 ± 1.0°; *p* = 0.034) and, in turn, the angle of separation in the latter was higher than in the batch at beginning of the ascending assessment (B2 = 9.6 ± 1.0°, B3 = 7.7 ± 1.2°; *p* = 0.003). No difference was found between batches at the beginning and at the end of the ascending assessment (B3 = 7.7 ± 1.2°, B4 = 7.4 ± 1.1°; *p* = 0.244)

Figure 4 shows the mean ± SD of latency to perform a choice after a sound production as a function of batch and type of trial.

Post hoc pairwise comparison between latency in training (Tr) and test trials (Te) within each batch showed significant difference in all batches (B1: Tr = 2.0 s, B1: Te = 2.4 s; *p* = 0.001; B2: Tr = 2.0 s, B2: Te = 2.3 s; *p* < 0.001; B3: Tr = 2.0 s, B3: Te = 2.3 s; *p* = 0.040; B4: Tr = 2.0 s, B4: Te = 2.2 s; *p* = 0.035). Regarding only test trials, the latency time in B1 was higher than in B4 (B1: Te = 2.4 s, B4: Te = 2.2 s; *p* = 0.002). No difference was found in latency time within the training trials among all batches. No difference was found in latency according of the type of reversal (success/failure). 

## 4. Discussion

The MAA of the dogs involved in this experiment ranged from 1.3° to 13.2°, with an average of 7.6°. The dog’s localization ability increased across the experiment, with an average MAA of 11.8° at the beginning of the entire procedure, to 7.4° at the end of it. Dogs’ improvement across the experiment was also evident by a slight decrement in their latency to choose, which went from 2.4 s to 2.2 s. 

The MAA found in dogs of our experiment had a wider range than that reported for dogs in the scientific literature [15,16], and, in most of our dogs, it fell below the previously reported minimum values (i.e., 7°, in the study by Babushina and Polyakov [15]). One evident element of difference between the present and the previous studies, which could have contributed to the different results, was our larger sample size. This likely allowed our results to reflect, more closely, the variability in the canine population. Yet, methodological differences may also have contributed to the detection of an extended range, especially in its lower end. Of particular relevance, the methodology adopted by previous studies did not seem to be perfectly suited to obtain stable (i.e., no further improvable by experience) and precise thresholds. Both previous studies used a modified method of constant stimuli [17], varying the angles of separations from large to narrow, and with pre-established values. Most of the resulting assessments were therefore performed on values which were far from the subject’s actual threshold. Conversely, the staircase procedure adopted in the present paper allowed us to concentrate the assessment close to the actual MAA of the subject, consolidating the final threshold with a large number of sessions around this value. Moreover, our termination criterion, i.e., that the six last sessions had to fall within a narrow range of 3°, prevented the possibility that our assessments ended whilst an improvement process was ongoing, and that the true final threshold had not yet been reached. It must also be considered that, with a minimum of about 600 to 1500 trials, the method of constant stimuli is very demanding in terms of number of trials required to identify the specific level of performance of a given dog. Comparatively, our procedure required about 120 sessions per dog. The advantages of using the staircase procedure over the method of constant stimuli has been pointed out before, in a study that compared the two methods in a psychophysics test in cats [22]. With the same number of trials, both methods were equally reliable but the staircase procedure led to a better approximation of the true thresholds which could be almost twice as small as the threshold reached with the method of constant stimuli. Hence, with significantly fewer trials we were able to allow for the improvement of the dogs’ performance until reaching a stable and reliable MAA.

It is interesting to compare the MAA of dogs in our study with those found in other mammals. Heffner and Heffner [23] studied the relationship between the MAA of 24 mammal species (including rodents, hoofed animals, aquatic animals, companion animals, monkeys and humans) and several ecological, anatomical and perceptual characteristics. The authors found that the strongest correlation was with the width of the field of best-vision (i.e., the retina area containing ganglion cell-densities at least 75% of maximum) (r = 0.911) which explains 83% of the variance in MAA among species. In particular, the narrower the field of best-vision, the better the sound-localization capability. For example, the field of best-vision was 132° wide in the horizontal plane in cows [24], 4.9° in cats [25] and around 0.75° in humans [26] and their MAAs were 30°, 5.7° and 1.3°, respectively. In dogs, the field of best vision was 5.1° wide ([27], cited in [23]), that is slightly higher than for cats, and dogs’ MAA found in the present study was 7.6°. Hence, our results are in agreement with the theory that sound-localization abilities is linked to the width of the field of best-vision proposed in this study. This relationship has a clear functional significance, which is linked to the species ecology and foraging behavior. Indeed, predator species typically have larger binocular fields [28], which allow them to precisely direct their eyes towards the source of a sound; thus, better sound and visual localization abilities are functional to predation. On the other hand, animals such as rodents and hoofed mammals, who have a narrow binocular field, and a wider broad retinal region with relatively high concentration of ganglion cells, do not need such a precise auditory localization of the sound source; as prey, they likely have a greater advantage in being able to spot the presence of a predator in a wider field. The high ability of dogs to locate the sound source could therefore depend on the fact that they are, or at least are descendants of, a predatory species. 

The inter-individual variability of MAAs in the present study was rather high, ranging from nearly 1° to nearly 13°. Albeit with all the limitations related to the low number of subjects in our sample, there are no indications that MAA was affected by individual differences in potentially relevant parameters, such as interaural distance or ear shape, nor by age or sex. 

Previous studies have proved that sound localization performance is negatively affected by age due to either asymmetrical hearing loss [29] or central age-related deficits [30]. Nevertheless, age-related deficits are unlikely to explain the variability observed in our sample. In fact, only dogs younger than 8 years were included, thus under the age at which age-related hearing [20,31] and cognitive [32,33] deficits are generally observed in dogs. Moreover, even though our youngest and oldest dogs had the lowest and the highest MAA, respectively, the absence of a clear age-related pattern of association between age and MAA in the overall sample suggests that age was not a significant factor. A characteristic aspect of the dog species that could be relevant in explaining the inter-individual differences in MAA, is the variability in the distribution of retinal ganglion cells among dogs [34], which is highly correlated with skull conformation (i.e., the length of the facial/nasal bones, relative to the skull’s width [35]). In line with the theory discussed above, it is possible that the difference in the distribution of retinal ganglion cells between the dolichocephalic dog, which present a wide retinal streak similar to that of hoofed animals, and brachiocephalic dogs, in which ganglion cells are concentrated in an area centralis similar to that of cats and humans, plays a role on sound-localization abilities at an individual or breed level. Although the MAA of the only clearly dolichocephalic dog in our sample (a whippet) fell close to the average MAA, we cannot exclude that with a larger sample size a relationship between cranial conformation and sound localization might be found.

One clear aspect that emerged in our procedure was that dogs showed an improvement throughout the assessments. This was supported by two parameters: first, the MAA of dogs improved across the entire procedure until the beginning of the ascending assessment, where the threshold was about 1.5 times lower than at the beginning of the descending assessment. The MAA did not further improve throughout the ascending assessment, indicating that the final threshold was already reached at the beginning of it. Second, the response speed (i.e., latency in choosing) in test trials decreased slightly, but significantly, from the beginning to the end of the procedure. This indicates that our dogs kept improving the speed of their choices from the first moment they were confronted with the task until beyond the moment in which they reached their maximum sensitivity. The observed improvement in the performance of our dogs can be easily explained by perceptual learning, which is defined as “any relatively permanent and consistent change in the perception of a stimulus array, following practice or experience with this array” [36]. This phenomenon is reported for discrimination of any kind of sensory cues such as tactile [37], visual [38], flavors [39] and acoustical stimuli [40]. Latency to react is linked to the difficulty of the task in humans [41], and the same phenomenon has been also reported in beagle dogs in several cognitive tasks [42]. Regardless, this underlines the ability of the present methodology to identify, with ease, the MAA achievable by dogs thanks to experience. It also highlights that to ensure that a methodology is adapted to the species and the task, the performance curve should reach a plateau to consider the estimated threshold as final. It should be noted that another possible contribution to the improvement observed in our procedure could be due to some inherent differential characteristic of the descending and ascending assessments and to the order by which they were administered. As we did not randomize the order by which the two assessments were administered to dogs, we do not know if and how starting the evaluation with the ascending assessment would have changed the performance in dogs. However, if some differences are to be expected for procedures in which a sensory threshold is approached from below or above the threshold itself, these go in the opposite direction of those observed here, i.e., worst performances are generally observed in ascending than in descending assessments [43]. Therefore, these differences cannot explain the improvement observed throughout our experiment. On the other hand, it was previously shown that in a difficult auditory-identification task, both humans and rats learnt better if they started with easy trials than if they completed only difficult trials throughout training [44]. In our case, the descending assessment was obviously easier in its initial trials, whereas the ascending one had a more stable (and harder) level of difficulty throughout. Therefore, our procedure seems to be the most suitable to perceive learning and obtain a stable threshold. In view of this, we believed that our procedure is a more suitable to obtain the best improvement in the MAA. 

## 5. Conclusions

Taken together, the results indicate that a staircase method is a feasible approach for the assessment of sound-localization thresholds in dogs. In particular, the method gave dogs enough repetitions to allow for a gradual improvement in performance through learning, and to reach a stable estimation of the MAA at the end of the procedure. Nonetheless, compared to previously used behavioral methods, the approach used in this study was also considerably faster.

The results also highlighted that the range of sound-localization thresholds in dogs is larger and potentially reaches much lower values, than previously reported. Although no clear patterns emerged from our experiment on the relationship between MAA and dog’s individual features, the results prompt the extension of the present study to larger-scale samples, in order to explore factors potentially explaining the inter-individual variability in sound localization abilities, including for instance breed or head and/or ear morphology.

## Figures and Tables

**Figure 1 vetsci-09-00619-f001:**
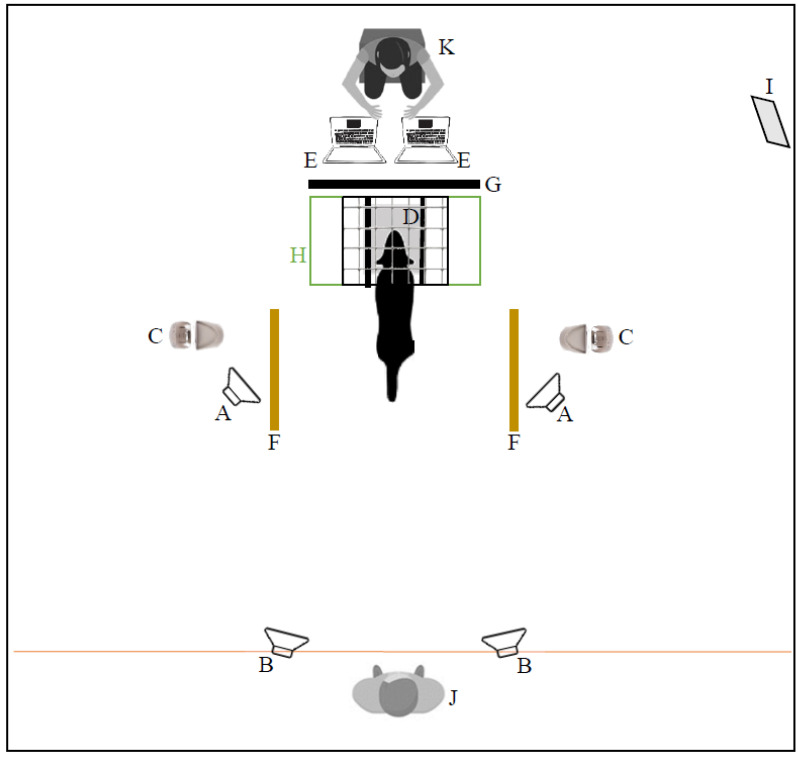
Schema of the room and spatial position of experimental elements from above. A, training speakers (fixed); B, testing speakers (movable); C, food dispensers; D, headrest; E, computers triggering sound; F, line defining dog’s choice; G, panel separating sound operator from dog’s view; H, apparatus; I, mirror; J, dog operator; K, sound operator.

**Figure 2 vetsci-09-00619-f002:**
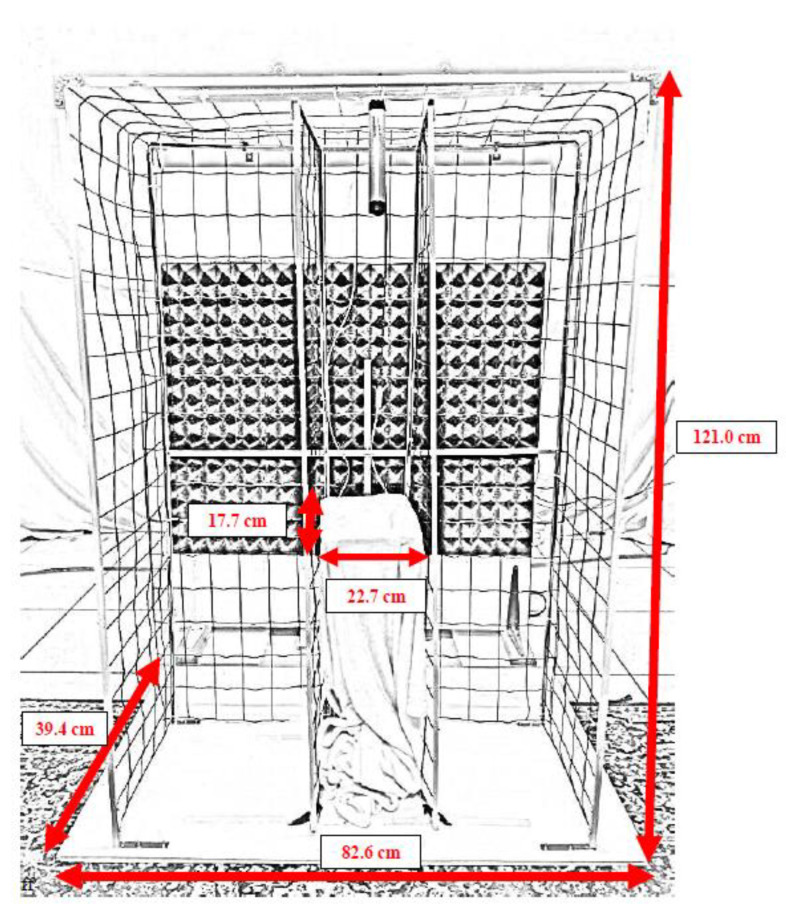
Detailed picture of the apparatus.

**Figure 3 vetsci-09-00619-f003:**
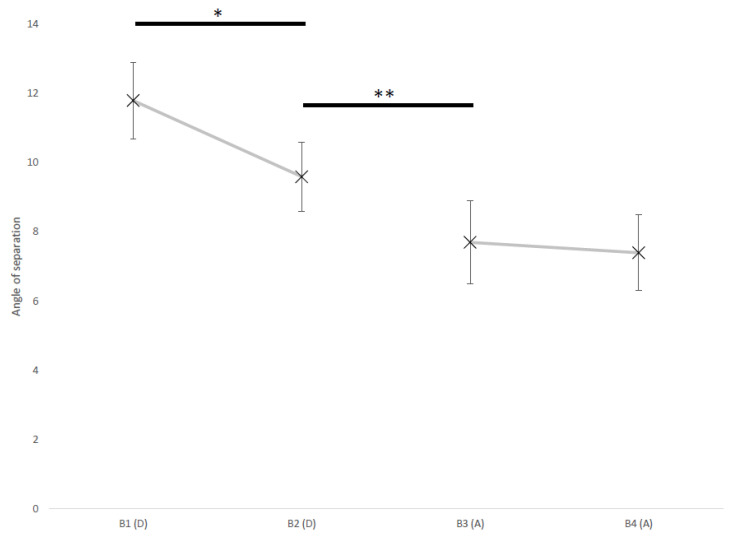
Mean ± SD of angle of separation in batches of four reversals (B1-B4), in the descending (D) and ascending (A) assessment. Significant differences of angle of separation between batches are flagged by asterisk(s) (* *p* < 0.05, ** *p* < 0.01, pairwise comparison after generalized estimating equations model).

**Figure 4 vetsci-09-00619-f004:**
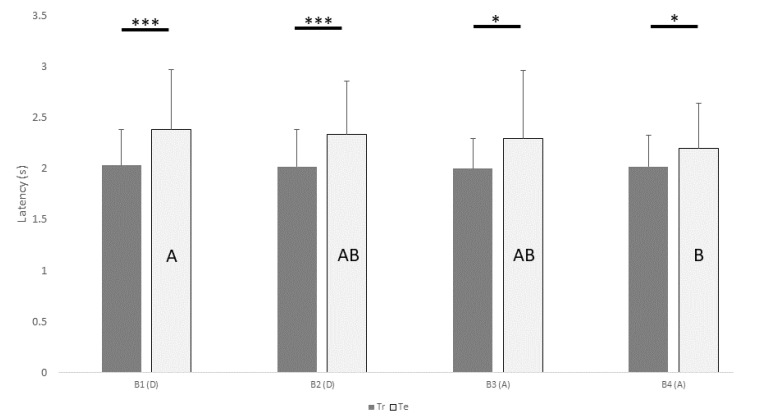
Mean ± SD of the latency from production of sound to the dog’s choice, as function of batch (B1-B4) and type of trial. Significant differences in means between training and test trials within the same batch are flagged by asterisk(s) (* *p* < 0.05, *** *p* ≤ 0.001). Significant differences in means in test trials between different batches are indicated by different capital letters (e.g. A indicates a significantly different mean compared to B, but not compared to AB), pairwise comparison after generalized estimating equations model). Tr, training; Te, testing; D, descending; A, ascending.

**Table 1 vetsci-09-00619-t001:** Demographics, ear shape and interaural distance of dogs.

Dog	Age (y)	Sex and Reproductive Status	Breed	Ear Shape	Interaural Distance (cm)
1	3.0	F/C	Mixed breed	Not covering ^1^	12.3
2	3.2	F/C	Golden retriever	Partially covering ^2^	12.2
3	1.2	F/I	Labrador	Partially covering ^2^	12.9
4	1.5	M/I	Australian shepherd	Not covering ^1^	15.0
5	3.5	F/C	Weimaraner	Partially covering ^2^	11.5
6	7.5	F/C	Mixed breed	Covering ^3^	10.0
7	6.8	F/C	Whippet	Not covering ^1^	8.5
8	1.4	M/C	Mixed breed	Not covering ^1^	12.9
9	7.8	F/C	Vizsla	Partially covering ^2^	10.5
10	3.6	M/I	Border collie	Partially covering ^2^	10.0

F, female; M, male; I, intact; C, castrated; ^1^ the ears were erect or semi-erect and are not covering at all the canal entrance; ^2^ the ears covered, without touching, the canal entrance; ^3^ the ears covered and touched the canal entrance.

## Data Availability

The data set analyzed for the current study is available from the corresponding author upon reasonable request.

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
