# Peer review of "Sound Localization Ability in Dogs"

_vetsci, 2022, doi:10.3390/vetsci9110619_

Round 1

Reviewer 1 Report

Sound localization ability in dogs - Review

Overall comments

Thank for you the opportunity to review this paper. It is an interesting and useful subject area with practical applications within canine training and management.

The paper is generally well written and constructed. Some minor language ‘blips’ that affect clarity and flow – I have added some suggested amends for areas where this might be considered valuable to improve and edit.

I liked the premise of the study and also how the study was undertaken - clear description of process and strategy.

Abstract

Clear and concise. Appropriate for study. Highlights key areas

Line 12-13 – reword ‘using methods that do not allow to’ – suggest ‘using methods that fail to’

Line 15 – ‘using angle of separation from 60° to 1°’ – suggest change ‘using’ to alternative as repeated

Line 16 – ‘allows to adapt the level of difficulty to the previous performance’ – suggest amend to ‘permits return to’ or similar for clarity

Line 25 – ‘the results prompt to inquire in larger scale studies’ – suggest reword to ‘the results suggest value in conducting larger scale studies’

Introduction

Clear with good definitions and clarity of background.

Sets the science for the study well and highlights current holes in knowledge relating to dogs. Nice consideration of existing work and critique of it.

Well written and to my knowledge, incorporates key studies and info that is relevant to this work.

Line 43-45 – a little unclear about the point about head movement or not – this might be worth expanding and clarifying…. have some species developed compensatory mechanisms to permit head movement and spatial hearing OR have they developed head movement to aid spatial hearing? This is especially relevant about how dogs ‘cock’ their heads based on sounds (see for example Sommese, A., Miklósi, Á., Pogány, Á. et al. An exploratory analysis of head-tilting in dogs. Anim Cogn 25, 701–705 (2022). https://doi.org/10.1007/s10071-021-01571-8)

Materials and Methods

Clear and well-structured for clarity

Line 99-100 - The age of the animals had to be between 1.0- and 7.9 year to exclude too lively young subject and animals with age-related hearing loss [18] – was this opportunistic sampling, but with a minimum and max age identified? Reword also for improved clarity – suggest ‘sample animals were aged between 1 and 8 years of age to minimise age effects of youthful exuberance and potential age-related hearing alterations’

Line 104 – reproductive status of dogs? Entire or not?

Line 107 – might be worth in main paper including details of breed/type/ear position/form rather than in supplementary material

Question – did dogs have training or acclimatisation to experimental protocol? How? How long for if so? – this is noted in section 2.4 but has no detail about length of training and any exclusions?

Line 145 – replace ‘capture’ with captured

Data analysis appears consistent and appropriate for study – any more clarification on specific behavioral data scored from videos relating to response to sound? (Perhaps nothing more was scored and that is OK!)

Results

Line 257 – I am a little unclear about the numbers here I am afraid relating to sessions. Could this be clarified somewhat? – ‘Dogs performed a total of 25.1 ± 4.5 and 19.2 ± 5.7 sessions, with 11.1 ± 1.6 and 10.8 ± 2.5 reversal sessions, in the descending and ascending assessments, respectively.’

Otherwise, results are clear and well presented

Discussion

Relevant and consistent discussion of findings.

Line 295-302 – could dog breed/type/size/ear position or conformation have an effect? – you do consider this in line 349, however. Perhaps further study might be relevant?!

Conclusion

Clear. Good review and reflection

Additional Info

Appears all appropriate and clear with good level of detail

References

Titled Reference?

I have not exhaustively gone through these, but there are occasional formatting errors and typos that need amended, including spacing and line positioning

Reviewer 2 Report

The study itself is very interesting. I have a few thoughts / questions about the study / article.

1. Spell check is needed as there are a few translation issues, grammatical changes needed and/or words missing

2. Was the dog operator someone known to the dogs or a stranger? Did part of the training include becoming familiar with the dog operator? Is it possible that the dog not feeling comfortable with the dog operator present might affect his behavior?

3. Your methodology is quite detailed, however, I am still having a hard time grasping exactly what occurred which might also impact replicability of the study. Not all of these may be relevant or known, but if so, it might be helpful. What was the 'sound?' Did the sound come at intervals or sporadically? Was it the same sound consistently? Did dogs eventually habituate to the sound after 120 sessions? How long was the dog required to stand in place? How was his welfare maintained? What constituted "success?" What if the dog's ears shifted in place but his head/body did not move? Were there differences in success based on time of day (e.g. upon  the start of the day or end of the day)?

4. Did the temperament of the dog matter? For example, if the dog was a more nervous dog, would that impact the dog's appetite or desire to take treats?

5. I am also curious if there was any significance with floppy eared dogs over pointy eared dogs. Did the dog breed matter? E.g. A German Shepherd Dog would seem to behave differently than a chihuahua or Shih Tzu. Do you have breed demographics for the study?

Thank you for your efforts and I am interested in seeing your study with a few more details included.

Round 2

Reviewer 2 Report

The changes you have made are excellent and have really enhanced your paper.